# The Dose-Response Associations of Sugar-Sweetened Beverage Intake with the Risk of Stroke, Depression, Cancer, and Cause-Specific Mortality: A Systematic Review and Meta-Analysis of Prospective Studies

**DOI:** 10.3390/nu14040777

**Published:** 2022-02-12

**Authors:** Yuanxin Wang, Renqing Zhao, Bin Wang, Chen Zhao, Baishu Zhu, Xin Tian

**Affiliations:** Department of Sports and Health Science, College of Physical Education, Yangzhou University, Yangzhou 225009, China; wangyuanxin880330@163.com (Y.W.); bwang1999@126.com (B.W.); zc072866@outlook.com (C.Z.); baishuzhu@outlook.com (B.Z.); tianxin96331@163.com (X.T.)

**Keywords:** sugar-sweetened beverage, stroke, depression, cancer, mortality

## Abstract

The associations between sugar-sweetened beverage (SSB) consumption and the risk of stroke, depression, cancer, and cause-specific mortality have not been determined, and the quantitative aspects of this link remain unclear. This meta-analysis therefore conducted a systematic review and dose-response analysis to determine their causal links. The database searches were conducted in PubMed, Cochrane library, Embase, Web of Science up to 10 November 2021. The intervention effects were evaluated by relative risk (RR) with 95% confidences (CI). Thirty-two articles met the inclusion criteria. Higher levels of SSB consumption significantly increased the risk of stroke (RR 1.12, 95% CI 1.03–1.23), depression (1.25, 1.11–1.41), cancer (1.10, 1.03–1.17), and all-cause mortality (1.08, 1.05–1.11) compared with none or lower SSB intake. The associations were dose-dependent, with per 250 mL increment of SSB intake daily increasing the risk of stroke, depression, cancer, and all-cause mortality by RR 1.09 (1.03–1.15), 1.08 (1.06–1.10), 1.17 (1.04–1.32), and 1.07 (1.03–1.11), respectively. The link was curved for depression and cancer risk (*p*_non-linear_ < 0.05). Subgroup analysis suggested that higher SSB intake increased ischemic stroke by 10%, CVD-caused mortality by 13%, and cancer-caused mortality by 6.0% than none or lower SSB consumption. It is suggested that SSB accounts for a leading risk factor of stroke, depression, cancer, and mortality, and that the risk rises in parallel with the increment of SSB intake (and is affected by participant characteristics).

## 1. Introduction

Although there is a decline in intake from sugar-sweetened beverages (SSB) in the US and Europe [1,2], the prevalence of SSB consumption worldwide remains very high [3]. SSB is considered as a leading source of energy intake. For example, SSB contributes about 9.3% of daily calories in men and 8.2% in women in US [4,5], and has been linked with weight gain, diabetes, hypertension, hyperlipidemia, gout, and coronary artery diseases (CAD) [6,7,8]. Presently, more and more people show their concerns about the effects of SSB on health. 

SSB intake has been recognized to induce hyperglycemia, dyslipidemia, inflammation, and endothelial dysfunction [9,10,11], which are assumed as risk factors for stroke. Recently, several studies reported higher SSB consumption increased stroke incidence [12,13]. Additionally, SSB intake often leads to obesity and type 2 diabetes mellitus (T2DM), which are considered as the risk factors of cancer [9,14,15], and evidence from observational studies suggested that there exited a close link between SSB and pancreatic cancer [16,17], breast cancer [17], colorectal cancer [18], and other cancers [19]. The SSB intake is also suspected to increase the risk of depression, and higher depression prevalence was observed among greater SSB drinkers both in prospective and cross-sectional studies [20,21,22]. Given the fact that SSB intake increases chronic diseases, such as stroke, cancer, diabetes, CVD, and hypertension, it subsequently increases mortality risk. It has been estimated that the consumption of SSB are associated with 184,000 deaths annually worldwide [23]. Recently, two large prospective studies observed a higher prevalence of all-cause mortality among greater SSB drinkers [24,25]. 

However, some important questions remain unaddressed yet. Firstly, the causal relation between SSB intake and chronic diseases has not been established because studies have reported inconsistent results on the associations of SSB with caner [16,26] or stroke [27,28]. Secondly, although previous meta-analyses reported the relations between SSB and risk of depression and cancer [29,30], the conclusions were drawn by incorporating case or cross-sectional studies, suggesting that the evidence was not substantial and convincing. Up to now, the cause-specific mortality risk related to SSB intake has not been determined [31], and new prospective studies have been reported since previous reviews [25,32,33]. Additionally, the quantified aspects of the associations between SSB intake and chronic diseases have not been determined yet. To address those concerns, we conducted a meta-analysis of cohort studies to determine the associations between SSB intake and the risk of stroke, depression, cancer, and mortality. The quantified aspects of this link and effects of the types of diseases and participant age, body mass index (BMI), total energy (TE) intake, location, and follow-up on the relation were also determined.

## 2. Materials and Methods

### 2.1. Search Strategies and Inclusion Criteria 

This systematic review and meta-analysis adhered to the preferred reporting items for systematic reviews and meta-analyses (PRISMA) guidelines [34]. Two authors, BZ and BW, conducted the electronic database searches in PubMed, Cochrane library, EMBASE, and Web of Science using a predetermined protocol up to 10 November 2021. After the removal of duplicates, the articles were screened on the basis of the titles and abstracts, and another author, CZ, reviewed them independently. The full-text of articles then were drawn and reviewed for eligibility independently by BZ and BW. To identify unpublished (ongoing or completed) studies, the WHO International Clinical Trials Registry Platform (ICTRP) search portal and ClinicalTrials.gov were searched up to 10 November 2021. The reference lists of relevant systematic reviews were screened manually to identify further potentially relevant citations. No language restriction was applied. The search strategies for electronic databases are listed in Appendix A.

The eligible studies should meet the following criteria: (1) prospective cohort studies; (2) the exposure was SSB intake, and the outcomes were incidents of stroke, depression, cancer, or mortality; (3) the participants were healthy adults at enrollment and aged ≥ 18 years; (4) for dose-response analysis, the levels of SSB consumption should be ranked at least three categories. We excluded case or cross-sectional studies, and studies conducted among children were also excluded. 

### 2.2. Data Extraction

Two authors (BZ and BW) independently extracted data from each included study and resolved any disagreement through discussion with another author (CZ). We contacted the corresponding authors to acquire the relevant information not reported in the original paper. The extracted details included the following items: the name of the first author; the study location; the source and the number of participants; mean or median age of participants; follow-up time; person-years, the events of stroke, depression, cancer and mortality, and adjusted relative risk (RR)/hazard ratio (HR) with 95% confidence (CI) for all categories of SSB consumption; covariates used for adjustment, etc. The data extraction followed the methods recommended by Cochrane Reviewers’ Handbook [35]. 

### 2.3. Risk of Bias Assessment

The quality of the included studies was evaluated independently by two authors (BZ and BW) using the Newcastle-Ottawa quality assessment scale assessment tools [36]. The results were discussed among the other authors (CZ and XT) for consensus. This scale assigned a maximum of nine points for each study. The following three broad perspectives are examined: the selection of cohorts (four points); comparability of cohorts (two points); ascertainment of the exposure and outcome of interest (three points). Finally, the score for each included study was listed in a table. 

### 2.4. Statistical Analyses 

RR with 95% CI were used to calculate the summary effects by the comparison of the risk of stroke, depression, cancer, and mortality between the highest and the lowest categories of SSB consumption. Fixed- or random-effects models were used according to between-study heterogeneity. The adjusted RR was used to conduct the meta-analysis. If only adjusted HR were given, the HR were designated as the RR. If studies did not report the person-years for each category, it was calculated by multiplying the total person-years by the percentage of populations in each category, or multiplying the number of participants in each category by follow-up years. For the included studies, the reported median or mean dose of SSB consumption in each category was designated as the SSB consumption category. 

SSB was considered as the main exposure. SSB was defined as any sweetened beverages, including soda, soft drinks, and sugar sweetened fruit juice, etc., not presented as diet. Doses of SSB reported as a serving/drink, ounce, can, and cup size were converted to milliliters per day for the dose–response meta-analyses. The conversions were in accordance with the definition in the original articles. For studies that did not define the portion size, it was converted by using recommended conversions (1 serving/drink = 250 mL, 1 can = 330 mL, and 1 cup = 200 mL).

Subgroup analysis was conducted to evaluate the intervention effects according to the types of diseases, and participants’ age (<60 years vs. ≥60 years), BMI (<24 kg/m^2^ vs. ≥24 kg/m^2^), TE intake (<1869 kcal/day vs. ≥1869 kcal/day), location (US vs. not US) and follow-up years (<10 years vs. ≥10 years). 

The dose-response meta-analysis was performed applying the methods described by Greenland and Longnecker [37], and the study-specific slopes (linear trends), and 95% CI were calculated from the natural logs of the RR and 95% CI across categories. The potential non-linear dose-response associations of diseases with SSB consumptions were estimated by using fractional polynomial models to build a four-knot restricted cubic spline regression.

The heterogeneity across studies was determined by Q value (with a significant level at *p* < 0.10) and I^2^ (low: I^2^ < 30%; moderate: 30% ≤ I^2^ < 60%; and high: I^2^ ≥ 60%). A two-tailed *p*-value < 0.05 was considered significant. All statistical analyses were performed using STATA software (Version 15, StataCorp LP, College Station, TX, USA). 

### 2.5. Sensitivity and Publication Bias Analyses

To conduct the sensitivity analysis, we determined the associations of SSB consumption with stroke, depression, cancer, and mortality risk by dropping single study and repeating the meta-analysis. To examine the publication bias or other sources of bias, funnel plots were constructed with treatment effects estimated from individual studies against a measure of study size (standard error of RR). Egger’s regression was used to explore the likelihood of the presence of small-study effects.

## 3. Results

### 3.1. Study Selection and Characteristics of Included Studies

The electronic database searches identified and screened 8839 abstracts, of which 8743 were excluded due to either duplicate study (n = 3351) or unrelated to the topic (n = 5392). Totally, 102 full-text articles were reviewed for eligibility and 70 were excluded with reasons: (1) not reporting data of interest (n = 10); (2) not prospective cohort studies (n = 25); (3) duplicate publication (n = 5); (4) not proper exposure (n = 19); and (5) others (n = 11). We identified unpublished information on 11 studies, but those registered studies did not meet eligibility criteria (Table 1). Finally, thirty-two studies were eligible for meta-analysis. (Figure 1, Table 1, Appendix A).

Details are presented in the characteristics of included studies (Table 1). Briefly, 32 reports from 28 cohorts included 3,505,329 participants and reported 13,485 stroke events, 14,166 cancer, 23,694 depression, and 99,126 death, with median follow-up of 14 years. The included studies were conducted in Australia (1 studies), US (17 studies), UK (2 studies), Japan (2 studies), Singapore (2 studies), France (1 studies), Spain (2 studies), Sweden (3 studies), and Europe (2 studies) (Table 1). 

### 3.2. SSB Consumption and Risk of Stroke

Seven cohorts generating eight study groups were included for the analysis of relationship between SSB intake and stroke [12,13,26,28,38,39]. The eligible studies included 350,684 participants and reported 13,483 stroke cases, with a median follow-up of 18 years and 97,903 person-years. Two types of stroke were reported, namely ischemic stroke and hemorrhagic stroke. The NOS score of individual studies was marked from 8 to 9 (Table 1). The heterogeneity between studies were low (I^2^ = 29.9%) (Appendix A). Summary analysis comparing the highest with the lowest categories of SSB consumption indicated that SSB intake significantly increased the risk of all-type stroke by 12% (RR 1.12, 95% CI 1.03–1.23) (Figure 2). We then re-analyzed effect estimates by stratifying stroke types and found that SSB consumption increased ischemic stroke by 10.0% (1.10, 1.01–1.20), but the estimates of effects were not significant for hemorrhagic stroke 6.0% (1.06, 0.92–1.22) (Figure 3). Subgroup analysis also suggested that the intervention effects tended to be significant for participants with age ≥ 60 years, BMI ≥ 24 kg/m^2^, TE ≥ 1869 kcal/day, living in US, and follow-up ≥ 10 years (Figure 3). Dose-response analysis demonstrated that each 250 mL increment of SSB intake daily increased the risk of all-type stroke by RR 1.09 (1.03–1.15) (Figure 4). The likelihood of the presence of publication bias was estimated by constructing funnel plots and examining Egger’s regression test. It is indicated that funnel plots are relatively symmetry (Appendix A), and there has weak evidence for the presence of small-study effects (Egger’s test, *p* = 0.495) (Appendix A). Sensitivity analysis indicated that the results were still significant after dropping a single study and re-examining effects estimates (Appendix A). 

### 3.3. SSB Consumption and Risk of Depression

Three cohorts generating 4four study groups were included for determining the associations of SSB consumption with depression [50,51,52]. The eligible studies included 287,556 participants and reported 23,694 depression cases, with a median follow-up of 10 years and 80,647 person-years. The NOS score of the included studies ranged from 8 to 9 (Table 1). The heterogeneity between studies were low (I^2^ = 0%) (Appendix A). Summary analysis indicated that participants with the highest SSB intake tended to have higher risk of depression (RR 1.25, 1.11–1.41) compared with the participants with the lowest SSB consumption (Figure 2). There existed a dose-response relationship between SSB and depression risk, with per 250 mL increment of SSB intake each day increasing the risk of depression by 8% (1.08, 1.06–1.10), and the link was curved (p_non-linear_ < 0.0001) (Figure 4). Subgroup analysis indicated that participants aged ≥ 60 years, BMI ≥ 24 kg/m^2^, TE < 1869 kcal/day, living in US, and follow-up < 10 years were more likely to develop depression for SSB intake (Figure 5). The possibility of presence of publication bias was determined by examining the funnel plots and conducting Egger’s test. The funnel plots were relatively symmetrical (Appendix A) and the evidence for the presence of small-study effects was weak (Egger’ test, *p* = 0.847) (Appendix A). Sensitivity analysis indicated that the results remained significant (Appendix A).

### 3.4. SSB Consumption and Risk of Cancer

Fifteen cohort studies were eligible for the analysis of relationship between SSB consumption and cancer [16,17,18,19,27,40,41,42,43,44,45,46,47,48,49]. The included studies enrolled 1,918,066 participants and reported 1416 cancer cases, with a median follow-up of 14 years and 677,267 person-years. Seven types of cancer were reported, i.e., pancreatic cancer, prostate cancer, glioma, endometrial cancer, breast cancer, colorectal cancer, and hepatocellular carcinoma. The NOS score of individual studies was marked from 8 to 9 (Table 1). The heterogeneity between studies were low (I^2^ = 0%) (Appendix A). Summary analysis showed that SSB consumption significantly increased the risk of all types of cancer by 10% (RR 1.10, 1.03–1.17) (Figure 2). The effect estimates were re-analyzed by stratifying cancer types and other participants’ characteristics finding that SSB consumption increased pancreatic cancer by 11.1% (1.11, 0.95–1.30), and all-type cancers for participants with any age (<60 years and ≥60 years), BMI (BMI < 24 kg/m^2^ and BMI ≥ 24 kg/m^2^), TE (TE < 1869 kcal/day and TE ≥ 1869 kcal/day), locations (US and not US), and follow-up (<10 years and ≥10 years) (Figure 6). Dose-response analysis demonstrated that per 250 mL increment of SSB intake daily increased the risk of all-type cancer by RR 1.17 (1.04–1.32), and the link was curved (p_non-linear_ < 0.05) (Figure 4). 

The likelihood of presence of publication bias was explored by examining funnel plots and analyzing Egger’s test. The funnel plots were somewhat asymmetry (Appendix A), and there seemed to exist in evidence for the presence of small-study effects (Egger’ test, *p* = 0.01) (Appendix A). It was suggested that two studies [27,44] contributed to the potential bias, and after dropping the two studies the estimation of Egger’ test was 0.058. We further examined the effects of the two studies on the intervention effects. The results indicated that the estimates effects were still significant (1.09, 1.03–1.16) after removal of the two studies, suggesting that the results were not affected by the two studies much. Sensitivity analysis was also performed by dropping a single study and re-examining the effects estimates. It is indicated that the results were robust (Appendix A).

### 3.5. SSB Consumption and Risk of Mortality

Eight prospective cohorts generating nine study groups were included for determining the associations of SSB intake with mortality [24,25,32,33,53,54,55,56]. The eligible studies included 949,023 participants and reported 99,126 deaths, with a median follow-up of 12.1 years and 913,357 person-years. There were two types of mortality, i.e., mortality caused by cardiovascular diseases (CVD) and cancer. The NOS score of the included studies was marked from 8 to 9 (Table 1). The heterogeneity between studies were high (I^2^ = 68.2%) (Appendix A) and random-effects methods were used for pooling effects estimates. Summary analysis indicated that SSB intake significantly increased the risk of all-cause mortality by 6% (RR 1.08, 1.05–1.11) (Figure 2). Subgroup analysis suggested that SSB intake increased both CVD- and cancer-caused mortality (RR 1.13, 1.06–1.20, and 1.06, 1.01–1.12, respectively), and increases all-cause mortality for participants with any age (<60 years and ≥60 years), BMI (BMI < 24 kg/m^2^ and BMI ≥ 24 kg/m^2^), TE (TE < 1869 kcal/day and TE ≥ 1869 kcal/day), locations (US and not US), and follow-up (<10 years and ≥10 years) (Figure 7). Dose-response analysis demonstrated that per 250 mL increment of SSB daily increased all-cause mortality risk by RR 1.07 (1.03–1.11) (Figure 4). 

We then examined the potential source of between-study heterogeneity and its effects on the results. It was suggested that one study [55] might contribute to the main source of heterogeneity since the score of I^2^ dropped to 47.7% after removal of this study and estimated effects were still significant (RR 1.14, 1.09–1.19). It is indicated that the inclusion of this study did not pose a threat on the stability of our results. The likelihood of presence of publication bias was evaluated by examining funnel plots and conducting Egger’s regression test. It was indicated that the funnel plots were relatively symmetry (Appendix A) and there had weak evidence for the presence of small-study effects (Egg’ test, *p* = 0.489) (Appendix A). Sensitivity analysis indicated that the results were robust (Appendix A).

## 4. Discussion

Our study has provided quantified evidence on the associations of SSB consumption with chronic diseases and death, finding that the risk of stroke, depression, cancer, and mortality increased in parallel to the increment of SSB intake. Additionally, the types of diseases and participant age, BMI, TE, location, and follow-up affected the detrimental relations. It is suggested that increasing SSB intake might increase the risks of the incidence of stroke, depression, cancer, and mortality in adults. 

The associations of SSB intake with the incidents of stroke have been well-investigated by observational studies [12,39]. However, it remains unclear which types of stroke are mostly affected by SSB consumption. Larsson et al. reported that SSB consumption significantly increased both ischemic and hemorrhagic stroke risk [39], whereas other studies found that the effects were only significant for ischemic stroke [12,38]. Our findings confirm that ischemic stroke is more sensitive to SSB intake. Additionally, the quantified aspects of the link between stroke risk and SSB intake remain undetermined. Our results demonstrated that the incidents of stroke increased in parallel to the doses of SSB consumption. There are several possible causal explanations for this observed association. SSB intake potentially increases metabolic syndrome and its components [6,57,58]. Findings from the randomized controlled trial observed a harmful effect of SSB consumption on low density lipoprotein (LDL) particles, fasting glucose, and high-sensitivity C-reactive protein (CRP) [58]. Another study found that SSB intake was positively associated with plasma triglycerides (TGs), IL-6, and tumor necrosis factor (TNF) receptors, and inversely associated with high density lipoprotein (HDL) cholesterol, lipoprotein, and leptin [57]. SSB intake also has been found to elevate blood glucose and insulin levels, and positively associated with weight gain and T2DM risk [6,59,60]. All of those adverse factors are well-established risk factors for stroke [61,62]. Those adverse factors also explained the findings that higher participant age, BMI, and TE tended to develop stroke in response to SSB consumptions. It is reported that SSB contributes a leading source of energy intake in US [4,5] indicating that Americans are prone to suffer from the adverse effects SSB consumptions on stroke. The longer follow-up indicated longer exposure of SSB intake for participants which might contribute to positive associations between SSB intake and stroke risk. 

The World Cancer Research Fund (WCRF) reports that overweight and obesity are linked to incidents of cancers, such as liver, advanced prostate, ovary, gallbladder, kidney, colorectum, esophagus, postmenopausal breast, pancreas, endometrium, and stomach cancers [63]. SSB intake is currently identified as a leading source for energy intake and a driver for obesity, and studies frequently observed positive associations between higher levels of SSB consumption and incidents of obesity-related cancers [15,45]. Additionally, a recent study reports that SSB intake increases the overall cancers (not obesity cancers only) [19], which indicates that besides obesity other risk factors, such as hyperglycemia, dyslipidemia, inflammation, and hyperinsulinemia [9,10,11], might contribute to this detrimental link. The results of subgroup analysis seemed to support those reports finding that any age, BMI, TE, location, and follow-up showed positive associations between SSB intake and cancers. 

Our study further suggests that the link is dose-dependent indicating that the risk of cancers increases in parallel to the amount of SSB intake. However, a recent meta-analysis found no association between SSB consumption and cancer risks [64]. This meta-analysis did not show the isolated associations in sugary drinks and artificially sweetened beverages, which might have impaired the possibility to detect a potential role of sugar drinks. Actually, a lack of correlation between artificially sweetened beverages and cancers has been frequently reported [15,40]. We did not find a close relation between SSB intake and pancreatic cancer. A previous meta-analysis also failed to find associations between SSB and pancreatic cancer risk [65]. Therefore, it is indicated that other types of cancer might contribute to the correlation between SSB and cancer risk.

The associations between SSB and the risk of cancer might be partly explained by their effect on overweight and obesity [66]. Additionally, SSB drinks also promote gains in visceral adiposity independently of body weight [66,67]. Visceral adiposity could promote tumorigenesis through alterations in adipokine secretion and cell signaling pathways [68]. Other pathways might relate to the high glycemic index or glycemic load of SSB consumption [69], which are recognized to be associated with hyperinsulinemia and type 2 diabetes [70].

The current meta-analysis for the first time provided quantified evidence on the associations between SSB and depression. Although SSB has been suspected of being related to depression risk, the causal link has not been established yet. Firstly, the associations between SSB and depression were frequently determined in cross-sectional studies [71,72]. Secondly, prospective studies have reported inconsistent results. For example, one prospective study within the National Institutes of Health American Association of Retired Persons (NIH–AARP) Diet and Health Study reported a detrimental correlation between SSB intake and depressive risk [50], whereas another cohort study indicated the correlation was not significant [73]. Our meta-analysis, based upon the evidence from prospective studies, ascertained a close link between SSB intake and depression risk. We further demonstrated the correlation is dose-dependent, suggesting that the more SSB is consumed, the more depression might be observed. There was one meta-analysis reporting on the relationship between SSB and depression, but the major evidence was based upon cross-sectional studies and the quantitative aspects of this link were not analyzed [29]. SSB consumption is associated with a variety of socio-economic and lifestyle factors and may contribute to obesity, diabetes, and poor health, which in turn contribute to the development of depression. Obesity has been found to be linked with increased cortisol production and higher hypothalamic–pituitary–adrenal (HPA) axis reactivity to psychological and physiological stress, which may lead to altered endocrine and stress responses [74].

In fact, consumption of added sugars from soft drinks has been linked with several metabolic disturbances such as impaired glucose homoeostasis and insulin resistance [75]. Low concentrations of brain derived neurotrophic factor (BDNF) have been recognized as facilitating neurogenesis and hippocampal atrophy in depression [76]. Rodents fed high-fat high-sugar foods rather than high-fat foods only have a reduction of BDNF levels [77,78], which could be a causal link between SSB consumption and depression. Additionally, sugar consumption has been associated with increased circulating inflammatory markers, which may depress mood [79,80]. Finally, high sugar intakes from soft drinks could induce hypoglycemia through a higher insulin response and thereby affects hormone secretion and potentially mood states [81]. Although the associations between SSB intake and depression seemed to be affected by participant age, BMI, TE, location, and follow-up, the limited studies included for subgroup analysis might not be sufficient to generate definitive conclusions. 

Recently, two large prospective studies (i.e., the Health Professionals Follow-up Study (HPFS) and Nurses’ Health Study (NHS) [24] and the European Prospective Investigation into Cancer and Nutrition (EPIC) [25]), reported that a higher level of consumption of SSB was associated with greater all-cause mortality in the United States and ten Europe countries, respectively. Since the consumption of SSB is associated with obesity [31,82], hypertension [83], T2DM [84], CVD [85], stroke [18], and cancer [86], all of which are considered as the leading causes of mortality. It has been estimated that the consumption of SSB were associated with 184, 000 deaths annually worldwide: 133,000 from diabetes, 45,000 from CVD, and 6450 from cancer [23]. In accordance with this report are the findings of the subgroup analysis showing that SSB intake increases the risk of CVD- and cancer- caused mortality, and the relationship is significant for higher BMI and any age, TE, and follow-up. Our findings further reveal a dose-dependent association between SSB consumptions and all-cause mortality indicating that the mortality risk elevated with the increment of SSB intake.

### The Strength and Limitations

One strength of the present meta-analysis is that only prospective studies were included, which is likely to draw a relative definitive conclusion. On the contrary, the cross-sectional or case studies offer less compelling evidence of possible causal links between SSB and chronic diseases due to the inevitable bias, for example selection bias. Furthermore, our study has determined quantified aspects of the relationship between SSB and the risk of stroke, depression, cancer, and mortality. Additionally, in comparison to previous meta-analysis, our review includes most recent well-designed prospective studies (for example, Anderson et al., 2020 [32]; Mullee et al., 2019 [25]; Zhang et al., 2020 [33]) which provided more valuable information about the relationship between SSB and chronic diseases. However, there exist some limitations of our study, for example for the analysis of the associations between SSB and depression, only three prospective studies were included, and subgroup analysis could be conducted only for pancreatic cancer due to the limited number of studies reporting other types of cancers.

## 5. Conclusions

Our study demonstrated that the risk of stroke, depression, cancer, and mortality increased in parallel to the increment of SSB consumption. The findings have clinical significance since the risk factors are modifiable, and reduction of SSB consumption helps to prevent some chronic diseases and disease-related mortality. However, well-designed prospective studies are still needed to confirm the findings of our reports and to determine the correlation between SSB consumption and the occurrence of some important chronic diseases, such as dementia and Parkinson’s disease.

## Figures and Tables

**Figure 1 nutrients-14-00777-f001:**
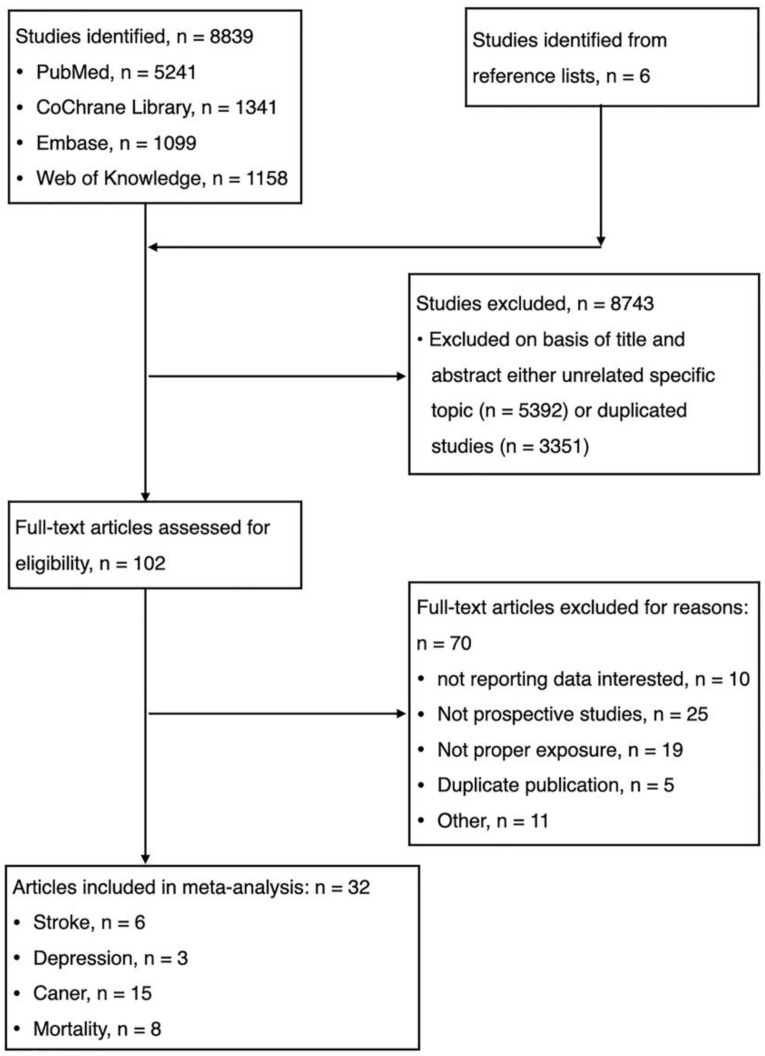
Flow chart for the selection of studies.

**Figure 2 nutrients-14-00777-f002:**
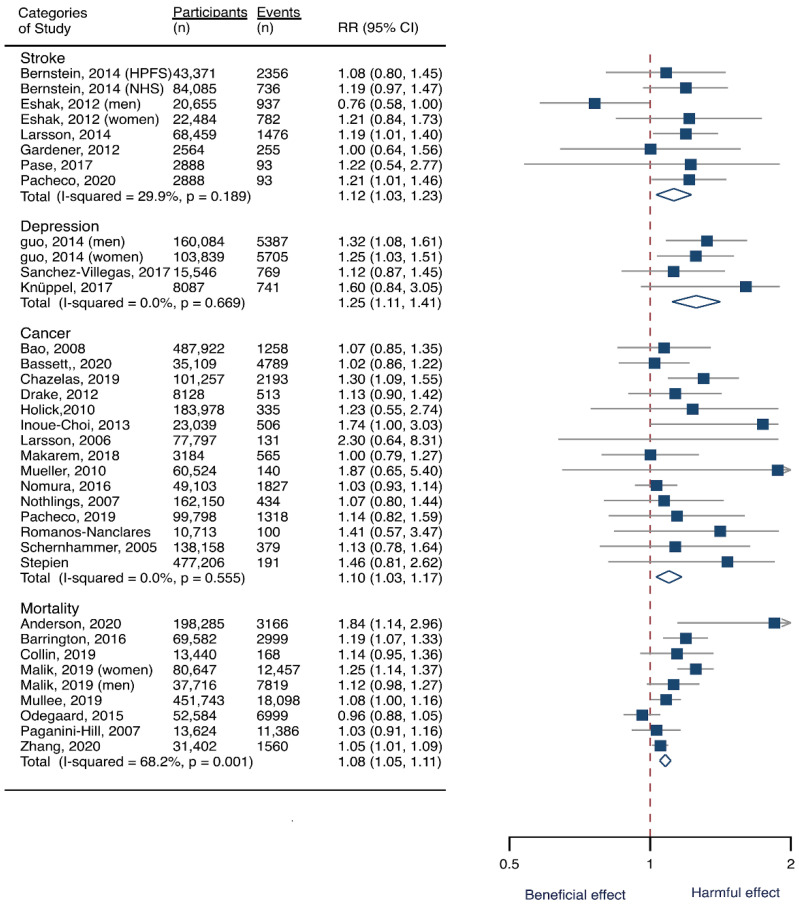
Overall analysis of the associations of sugar-sweetened beverage (SSB) intake with the risk of stroke, depression, cancer, and mortality. The square indicates relative risk (RR) with 95% confidence interval (CI). The rhombus denotes effects sizes.

**Figure 3 nutrients-14-00777-f003:**
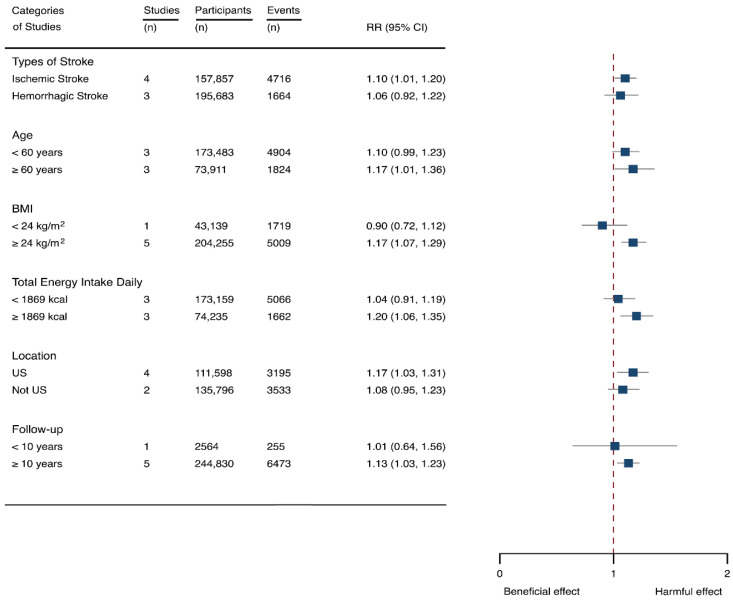
Subgroup analysis of the associations between sugar-sweetened beverage (SSB) intake and stroke stratified by the types of stroke, age, body mass index (BMI), energy intake daily, location, and follow-up years. The square indicates relative risk (RR) with 95% confidence interval (CI).

**Figure 4 nutrients-14-00777-f004:**
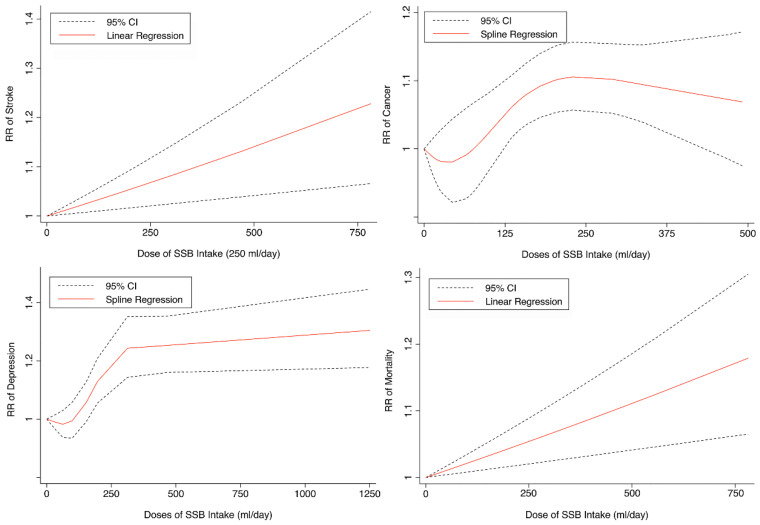
Linear or spline regression of the link between sugar-sweetened beverage (SSB) intake and the risk of stroke, depression, cancer, and mortality. RR: relative risk. CI: confidence interval.

**Figure 5 nutrients-14-00777-f005:**
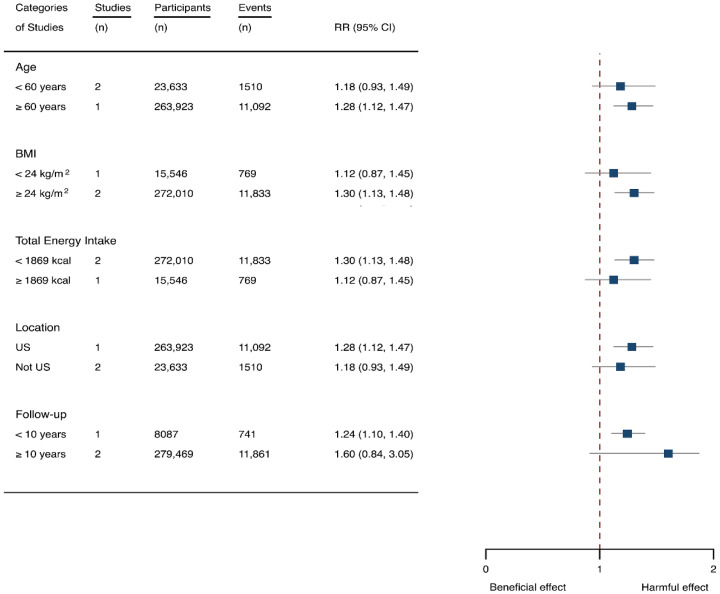
Subgroup analysis of the associations between sugar-sweetened beverage (SSB) intake and depression stratified by age, body mass index (BMI), energy intake daily, location, and follow-up years. The square indicates relative risk (RR) with 95% confidence interval (CI).

**Figure 6 nutrients-14-00777-f006:**
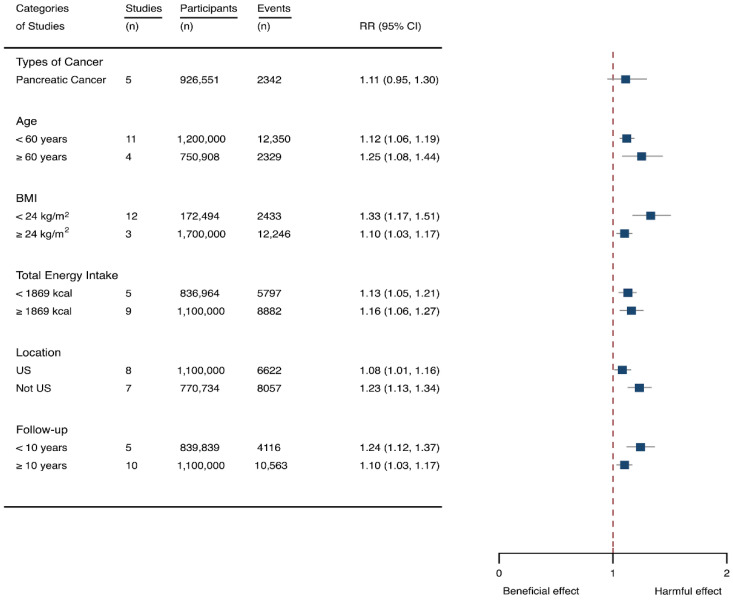
Subgroup analysis of the associations between sugar-sweetened beverage (SSB) intake and caner stratified by the types of cancer, age, body mass index (BMI), energy intake daily, location, and follow-up years. The square indicates relative risk (RR) with 95% confidence interval (CI).

**Figure 7 nutrients-14-00777-f007:**
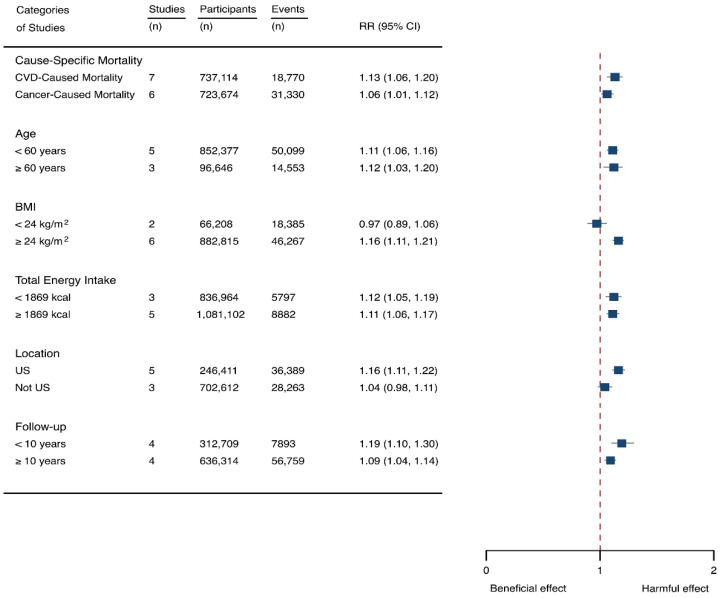
Subgroup analysis of the associations between sugar-sweetened beverage (SSB) and mortality stratified by cause-specific mortality, age, body mass index (BMI), energy intake daily, location, and follow-up years. The square indicates relative risk (RR) with 95% confidence interval (CI).

**Table 1 nutrients-14-00777-t001:** Characteristics of included studies.

Authors	Countries	Cohorts	Follow-Up (yrs)	Participants	Age (yrs)	Outcomes	Events (n)	QS
Stroke								
Bernstein, 2012 [12]	US	HPFS & NHS	28/22	43,371 men and 84,085 women	40–75 /30–55	Ischemic and hemorrhagic stroke	2356; 736	8
Eshak, 2012 [38]	Japan	Prospective study	18	43,139 men and women	40–59	Ischemic and hemorrhagic stroke	1047; 672	9
Larsson, 2014 [39]	Sweden	Prospective study	10.3	68,459 adults	45–83	Ischemic and hemorrhagic stroke	1220; 256	9
Gardener, 2012 [26]	US	NOMAS	9.8	2564 adults	69	Stroke	255	6
Pacheco, 2020 [28]	US	CTS	20	106,178 women	52	Stroke	5258	8
Pase, 2017 [13]	US	Prospective study	10	2888 Adults	>45	Ischemic stroke	93	9
Cancers								
Bao, 2008 [16]	US	NIH-AARP	7.2	487,922 men and women	50–71	Pancreatic cancer	1258	9
Bassett, 2020 [19]	Australia	MCCS	19	35,109 men and women	27–76	All cancers	4789	8
Chazelas, 2019 [40]	France	NutriNet-Santee	5.1	101,257 adults	42	All cancers	2193	9
Drake, 2012 [41]	Sweden	MDC	15	8128 men	45–73	Prostate cancer	513	9
Holick, 2010 [42]	US	Prospective study	14–24	183,978 adults	36–55	Glioma	335	8
Inoue-Choi, 2013 [43]	US	IWHS	14	23,039 postmenopausal women	52–71	Endometrial cancers	506	8
Larsson, 2006 [44]	Sweden	SMC & COSM	7.2	77,797 women and men	45–83	Pancreatic cancer	131	9
Makarem, 2018 [45]	US	FHS	20	3184 adults	26–84	All cancers	565	8
Mueller, 2010 [27]	Singapore	Prospective study	14	60,524 adults	45–74	Pancreatic cancer	140	8
Nomura, 2016 [46]	US	BWHS	16	49,103 Black Women	21–69	Breast cancer	1827	8
Nothlings, 2007 [47]	US	Prospective study	8	162,150 adults	45–75	Pancreatic cancer	434	9
Pacheco, 2019 [18]	US	Prospective study	20	99,798 female teachers and administrators	49–56	Colorectal cancer	1318	8
Romanos-Nanclares, 2019 [17]	Spain	SUN	2	10,713 Spanish females	33	Breast cancer	100	8
Schernhammer, 2005 [48]	US	NHS & HPFS	20	138,158 men and women	30–55	Pancreatic cancer	379	9
Stepien, 2016 [49]	Europe	EPIC	11.4	477,206 adults	51–60	Hepatocellular carcinoma	191	9
Depression								
Guo, 2014 [50]	US	NIH-AARP	11	263,923 Adults	50–71	Depression	11,092	9
Knüppel, 2017 [51]	UK	The Whitehall Study II	5	8087 adults	35–55	Depression	741	9
Sanchez-Villegas, 2017 [52]	Spain	SUN	10	15,546 Spanish university graduates	25–65	Depression	769	8
Mortality								
Anderson, 2020 [32]	UK	Prospective study	7	198,285 men and women	40–69	All-cause mortality	3166	9
Barrington, 2016 [53]	US	VITAL	5	69,582 adults	50–76	CVD- and cancer-caused mortality	1066; 1933	9
Collin, 2019 [54]	US	REGARDS	6	13,440 adults	64	CVD-caused mortality	168	9
Malik, 2019 [24]	US	HPFS	34	118,363 men and women	40–75	CVD- and cancer-caused mortality	7896; 12,380	9
Mullee, 2019 [25]	Europe	EPIC	16.4	451,743 adults	51	CVD- and cancer-caused mortality	5867; 12,231	9
Odegaard, 2015 [55]	Singapore	SCHS	16.3	52,584 Chinese men and women	45–74	CVD- and cancer-caused mortality	3097; 3902	9
Paganini-Hill, 2007 [56]	US	Prospective study	23	13,624 men and women	74	All-cause mortality	11,386	8
Zhang, 2020 [33]	US	NHANES	7.9	31,402 adults	≥20	CVD- and cancer-caused mortality	676; 884	8

Notes: The quality of the included studies was evaluated using the Newcastle-Ottawa quality assessment scale assessment tools. yrs, years; QS, quality score; CVD, cardiovascular diseases.

## Data Availability

All data are reported in this manuscript.

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
