# Peer review of "The Dose-Response Associations of Sugar-Sweetened Beverage Intake with the Risk of Stroke, Depression, Cancer, and Cause-Specific Mortality: A Systematic Review and Meta-Analysis of Prospective Studies"

_nutrients, 2022, doi:10.3390/nu14040777_

Round 1
Reviewer 1 Report
Wang, Zhao Wang, et al: Dose-response of sugar-sweetened beverage intake.
Abstract: acceptable in current form and does represent focus of paper.
Introduction:
I really question the first line of the paper, please review and perhaps consider how to revise this statement because it is not completely universal. See: Vercammen KA, Moran AJ, Soto MJ, Kennedy-Shaffer L, Bleich SN. Decreasing Trends in Heavy Sugar-Sweetened Beverage Consumption in the United States, 2003 to 2016. J Acad Nutr Diet. 2020 Dec;120(12):1974-1985.e5.
L45: Remove extra space, all studies have flaws (controversial) authors would improve their paper if they focused on what the specific flaws/controversy for ref 16, 26-28 were.
L50: incomplete sentence…though there had….
L58: Authors would greatly improve the unique nature of their paper if they improved content-focus of Lines 57-59
Overall, the content of the introduction is “ok” but just does not present an over whelming reason for why the present study is needed.
Materials and Methods:
I like how authors addressed Risk of bias assessment in section 2.3….more authors should include this in their work
Section 2.4: Try not to start so many paragraphs with “WE”, otherwise section written nicely.
Otherwise acceptable and makes it possible for others to understand how study was conducted.
Another unusual question is related to filters on internet content placed on information collected from the web by the Government of China. My understanding is that there are filters that limit access to some content “could” this have been applied to your PubMed, Cochrane, etc screening searches without the investigators knowledge? Might be worth considering this.
Results:
Figure 2. Need to capitalize O in “overall”, also authors might wish to ponder wording and meaning of “favor”. Being more likely to have an increase incidence of stroke or mortality would not be “favorable” in my view, though these events are more “probable” in those consuming more SBB.
L176: “we explored the likelihood…” thanks for explaining your though process and rational for flow of ideas.
L204: I think you should be saying “…relatively symetical…” not symmetry
Figure 4: authors may wish to remind reader of why two insets were “Spline” regressions and two were “Linear” Regressions. ALSO again you need to always capitalize the first letter in the figure legend.
I did very much like the authors use of dose-response with respect to risk and SSB intake
Discussion:
L279: I am not sure that the data suggests causality in the sense that Reducing Intake will lead to reduced incidence I the future. I would agree that the data does suggest that those with a history of SSB use do certainly present with greater risk in the future.
L289: I like your willingness to address :”Causes”
L333: personally I might suggested making SSB-Depression more of a focus of paper and especially the discussion in order to better demonstrate why the findings of your Meta-Analysis is unique and needed.
L382: Authors really ought to just include the years and studies what had not been included in the prior studies to remind the reader of why their paper is more valuable.
L393: incomplete sentence- “…we still need further study to….”
Overall Comments: Authors put a lot of work into this paper, but I am just not convinced that it adds dramatically to our understanding of the link between SSB intake and Health, although SSB dose- Health response data presented are useful.
Author Response
Response to reviewer #1:
Many thanks for the review of our manuscript. The comments have provided valuable suggestions for the improvement of the quality of our manuscript. We have carefully read the comments and made a point-by-point revise. The language of the manuscript has been checked carefully.
Reviewer 1
Introduction:
Point 1: I really question the first line of the paper, please review and perhaps consider how to revise this statement because it is not completely universal. See: Vercammen KA, Moran AJ, Soto MJ, Kennedy-Shaffer L, Bleich SN. Decreasing Trends in Heavy Sugar-Sweetened Beverage Consumption in the United States, 2003 to 2016. J Acad Nutr Diet. 2020 Dec;120(12):1974-1985.e5.
Response 1: We have revised this sentence according to the comments.
Point 2: L45: Remove extra space, all studies have flaws (controversial) authors would improve their paper if they focused on what the specific flaws/controversy for ref 16, 26-28 were.
Response 2: Extra space has been removed. We have revised this sentence to point out that studies on the associations of SSB with cancer or stroke reported controversial results.
Point 3: L50: incomplete sentence…though there had….
Response 3: We have modified this sentence according to the comments.
Point 4: L58: Authors would greatly improve the unique nature of their paper if they improved content-focus of Lines 57-59
Response 4: Thanks for the suggestion. We have reworded this sentence to focus on the unique nature of this paper.
Overall, the content of the introduction is “ok” but just does not present an over whelming reason for why the present study is needed.
Materials and Methods:
Point 5: I like how authors addressed Risk of bias assessment in section 2.3….more authors should include this in their work
Response 5: We have addressed more details of Risk of Bias Assessment.
Point 6: Section 2.4: Try not to start so many paragraphs with “WE”, otherwise section written nicely.
Otherwise acceptable and makes it possible for others to understand how study was conducted.
Response 6: The first sentence of each paragraph in this section was carefully checked and reworded not to start with “WE”. We also modified this section to improve the quality.
Point 7: Another unusual question is related to filters on internet content placed on information collected from the web by the Government of China. My understanding is that there are filters that limit access to some content “could” this have been applied to your PubMed, Cochrane, etc screening searches without the investigators knowledge? Might be worth considering this.
Response 7: The comments are important for conducting electronic database searches. The access to academic database such as Elsevier, Springer Nature, PubMed, Cochrane Library, etc. is not limited. And the database searches can be applied to PubMed, EMBASE, Cochrane Library, etc.
Results:
Point 8: Figure 2. Need to capitalize O in “overall”, also authors might wish to ponder wording and meaning of “favor”. Being more likely to have an increase incidence of stroke or mortality would not be “favorable” in my view, though these events are more “probable” in those consuming more SBB.
Response 8: The O in “overall” has been capitalized. We have replaced the word “favor” with “Beneficial effect/Harmful effect” in Figures according to the comments.
L176: “we explored the likelihood…” thanks for explaining your though process and rational for flow of ideas.
Point 9: L204: I think you should be saying “…relatively symetical…” not symmetry
Response 9: We have revised the word “symmetry”.
Point 10: Figure 4: authors may wish to remind reader of why two insets were “Spline” regressions and two were “Linear” Regressions. ALSO again you need to always capitalize the first letter in the figure legend.
Response 10: We have capitalized the first letter in the figure legend according to the comments.
I did very much like the authors use of dose-response with respect to risk and SSB intake
Discussion:
Point 11: L279: I am not sure that the data suggests causality in the sense that Reducing Intake will lead to reduced incidence I the future. I would agree that the data does suggest that those with a history of SSB use do certainly present with greater risk in the future.
Response 11: This sentence has been revised according to the suggestion.
Point 12: L289: I like your willingness to address :”Causes”
Response 12: This sentence has been reworded as “there are several explanations for the causes of this link”.
Point 13: L333: personally I might suggested making SSB-Depression more of a focus of paper and especially the discussion in order to better demonstrate why the findings of your Meta-Analysis is unique and needed.
Response 13: We have rewritten this paragraph according to the suggestion.
Point 14: L382: Authors really ought to just include the years and studies what had not been included in the prior studies to remind the reader of why their paper is more valuable.
Response 14: We have listed the years and studies that were not included in previous meta-analysis.
Point 15: L393: incomplete sentence- “…we still need further study to….”
Response 15: This sentence has been modified.
Overall Comments: Authors put a lot of work into this paper, but I am just not convinced that it adds dramatically to our understanding of the link between SSB intake and Health, although SSB dose- Health response data presented are useful.
Reviewer 2 Report
Review methodology was adequate, write up was easy to follow. Check with the Journal about the use of We throughout.
Author Response
Response to the comments of reviewer #2:
Many thanks for the review of our manuscript. The comments are helpful for improving the quality of our manuscript. The language of the manuscript has been checked carefully and we have modified the use of “We” to meet the requirement of the Journal.